# Integrative Interventions for Improving Outcomes in Depression: A Narrative Review

Matthew Halma [1,*] , Christof Plothe [1] and Paul E. Marik [2,*]

1   EbMC Squared CIC, Bath BA2 4BL, UK; plothe@ostmed.net
2   Frontline COVID-19 Critical Care Alliance, Washington, DC 20036, USA
*   Correspondence: matt@worldcouncilforhealth.org (M.H.); pmarik@flccc.net (P.E.M.)

**Abstract:** Antidepressants are among the most used medications in the US, with significant deleterious effects on people's well-being. At any given time, depression impacts approximately 1 in 10 Americans, causing wide and broad societal costs. Interest is developing for non-pharmacological treatments and preventative measures. We summarize the literature on non-invasive dietary and lifestyle approaches for treating depression. This review aims to inform future research and treatment programs for depression by providing an evidentiary summary of integrative therapeutic approaches for depression.

**Keywords:** depression; antidepressants; integrative medicine; social determinants of health; public health

## 1. Introduction

Worldwide, the prevalence of depression among adults is estimated to be 28.4% [1]. Among adolescents, the one-year prevalence of depression is 8% and the lifetime prevalence of depression is 19% [2]. Among adolescents, the prevalence of elevated depressive symptoms rose by nearly 50% between the decade of the 2000s and the decade of the 2010s [2]. Depression also affects healthcare workers and is more common among medical professionals than in the general population [3,4].

Beyond the immediate effects of depression, those with depression are at higher risk for many other conditions, including heart attacks [5], diabetes [6], and suicide [7]. The debilitating nature of the condition makes it difficult for one to enjoy a fulfilling life with social connections [8]. Depression has many multifarious impacts on one's career prospects and one's ability to experience joy from goal pursuit or enacting a hobby [9].

Depression can also affect others besides the depressed person; depressed people often withdraw from social relationships [10], and those they do still meet with can be influenced through mental contagion [11,12].

In the Netherlands, about 1 in every 13 adults is currently using an antidepressant [13]. In recent years, the trend has been towards the increasing duration of antidepressant use [14], and in the US two-thirds of patients continue antidepressants for at least two years [15,16].

## 2. Epidemiology

While depression is seen as a condition affecting solely the mind, still epidemiological factors underlie depression at the population level. Depression is more common in women [17], and peaks in the 45 to 59 years age group [18]. Rural residents also have lower rates of depression compared to urban dwellers [19]. Those involved in some spiritual practice reduce the risk of developing depression by half [20].

Characteristics of one's family of origin can predispose one to depression. While the mechanism of causation is unclear, the children of parents with major depressive disorder (MDD) are three times more likely to develop MDD than the children of parents without MDD [21]. A history of childhood abuse is associated with a greater risk of

depression [22,23], as is a lack of parental affection [24]. Social contagion of moods is possible, and being around other depressed people may contribute to depression [11].

Other factors associated with depression are low education, recent negative life events, loneliness, alcohol consumption, low physical activity [25,26], and smoking [27]. The personality traits of low agreeableness, low extraversion, low openness, low mastery, low conscientiousness, and high neuroticism are also associated with depression [27]. Internet addiction is also associated with a greater depression risk [28]. Significant life disruptions can also precipitate the onset of depression, including heart attacks [29] and even childbirth [30]. Due to the mental nature of depression, a discussion about its causes traverses many questions about how one is living one's life, including work [31,32], relationships, and self-care.

Nutritional associations have also been investigated. Coffee consumption was associated with a lower risk of depression in women [33]. Vegetable and fruit consumption was also associated with a lower risk of depression [34], and dietary magnesium and calcium significantly lowered the depression risk [35]. A recent umbrella review of the dietary associations with depression prevention and treatment demonstrated a significant protective benefit from healthy diet patterns [36]. Unhealthy beverage consumption habits, as parametrized by the Healthy Beverage Index (HBI) score [37], were also associated with an increased depression risk [38].

Specific factors showing strong evidence for decreased depression risk included fish consumption, coffee or tea consumption, dietary zinc, and light to moderate alcohol (<40 g/day) [36]. Consumption of sugar-sweetened beverages also raised the risk of depression [36]. Moderate-quality evidence exists for the association of consumption of probiotics, omega-3 polyunsaturated fatty acid, and acetyl-L-carnitine with decreased depression risk [36].

Other dietary factors studied for their anti-depression effect, revealing equivocal evidence for efficacy, include cocoa-rich foods [39], red or processed meat [40], vitamin D [41], folic acid [42], and B vitamins [43]. Genetic factors can contribute to or protect against the depression phenotype [44–47], and heritability of depression is estimated at 37% from twin studies [48].

Exposure to nature is associated with better affective states and lower risk of depression [49–52], an effect which can be mediated by the quality of the urban built environment [53,54]. Sociality can also protect against depression, as evidenced by an interventional study, where women with chronic depression in London made new friends, and observed a significant improvement in their present state examination (PSE) scores [55], an assessment of effect [56]. Additionally, having hobbies is associated with a lower risk of depression [57,58].

High-stress environments and jobs can also contribute to depression, though this relationship is mediated by other factors [59], including (perceived) level of support [60] and psychosocial safety in the workplace [61].

### 3. Aims and Methods

This narrative review aims to include non-pharmacological, evidence-based treatments for the treatment of depression. We divide these into two broad categories of interventions: dietary/nutraceutical and lifestyle therapies.

The methodology of this article begins with first performing a manual search for (1) dietary factors impacting depression, including specific supplements and herbal treatments, and (2) lifestyle factors influencing depression. The search strategy searches for reviews summarizing the interventions in each category. When reviews are found, the interventions summarized in the review are included in Supplementary Table S1 for dietary interventions with clinical evidence. Agents with only preclinical evidence are included in Supplementary Table S2. Lifestyle interventions are included in Supplementary Table S3. The dietary interventions with human clinical evidence for depression treatment are in-

cluded in Section 4. The lifestyle interventions with human clinical evidence are included in Section 5.

## 4. Nutritional Support for Depression Treatment

As mentioned, the nutritional factors behind depression have been elucidated in meta-analyses. Vegetarian diets are associated with a higher rate of depression in people [62], whereas Mediterranean diets are associated with a lower risk of depression [63,64]. Other specific nutrient deficiencies and their impact on depression are outlined in Table 1.

Nutraceuticals in the context of depression have been reviewed in [42,65–73]. Several nutritional deficiencies may exist in the depressed patient [74], which, if addressed, may positively influence the prognosis of depression.

We searched for reviews on nutritional supplements in depression and found several reviews providing an evidentiary overview of different nutraceutical and nutritional supplementary protocols for depression (Supplementary Table S1) [42,65–70,75–82]. The specific interventions are included in Table 1.

**Table 1.** A summary of dietary agents and their impacts on depression.

| Factor | Impact | Optimal Serum Levels | Daily Intake in Depression Treating Context | Sources |
|---|---|---|---|---|
| Zinc | Zinc supplementation significantly lowered depressive symptom scores (Beck's Depression Inventory, BDI) WMD = −4.15; [−6.56, −1.75] [83] | 70–120 micrograms per deciliter (mcg/dL) for adults | 25 mg zinc sulfate or 30 mg zinc gluconate [83] | Meat, shellfish, dairy, legumes, nuts |
| Magnesium | Consumption associated with lower risk of depression RR = 0.81 [0.70, 0.92] [35] | 0.75–0.95 millimoles per liter (mmol/L) for adults | 248 mg [84] | Leafy greens, nuts and seeds, legumes |
| Caffeine | Associated with reduced depression risk RR = 0.72 [0.52, 1.00] highest vs. lowest consumption [85] | N/A | Between 68 mg/day and 509 mg/day [85] | Coffee, tea |
| Cocoa | Decrease in depressive symptoms g = −0.42 [−0.67, −0.17] [39] | N/A | 50–100 g/day cocoa | Cocoa |
| Fish | Lowers depression risk RR = 0.89 [0.80, 0.99] highest vs. lowest consumption [86] RR = 0.83 [0.74, 0.93] highest vs. lowest consumption [87] | N/A | >1 serving per week [86] | Fish |
| Omega 3 polyunsaturated fatty acids | Lowered depression risk RR = 0.87 [0.74, 1.04] highest vs. lowest consumption [86] EPA + DHA consumption associated with lower depression risk [88] | N/A | 500 mg/day [86] | Fatty Fish |

**Table 1.** *Cont.*

| Factor | Impact | Optimal Serum Levels | Daily Intake in Depression Treating Context | Sources |
|---|---|---|---|---|
| Selenium | Intake associated with lower risk of postpartum depression OR = 0.97 [0.95, 0.99] and reduction in depressive symptoms WMD = −0.37 [−0.56, −0.18] [89] | Average level 124 ng/mL [90] | 100 to 200 µg [89] | Wheat products, meat [91] |
| B-vitamins | Non-significant reduction in depressive symptoms (SMD = 0.15 [−0.01, 0.32]) [43] | N/A | N/A | Liver, fish, leafy greens, eggs, seeds |
| Biotin | Associated with lower odds of depression (OR = 0.71 [0.55, 0.91]) [92] | >400 ng/L [93] | 30 µg [94] | Organ meat, egg yolk, some vegetables, milk [95] |
| Folic acid | Associated with lower odds of depression OR = 0.78 [0.61, 0.99] [92] | Deficiency is defined as serum folate < 10 nmol/L and RBC folate < 340 nmol/L [96] | 240 µg [94] | Legumes, leafy greens, citrus, vegetables, liver, dairy products [97] |
| Vitamin B12 | No significant effect on depressive symptoms [98] | Deficiency is defined as plasma vitamin B12 < 150 pmol/L [96] | 2.4 µg [94] | Liver, fish, leafy greens, eggs, seeds |
| Vitamin D | In cases of deficiency, vitamin D supplementation may help depressive symptoms [99] Inverse correlation between serum vitamin D levels and depression [100] | Serum 25-Hydroxyvitamin D: 50–100nmol/L [101] | >1000 IU [99] | Sunlight [102], oily fish, fortified foods [103] |
| Probiotics | Small but significant effects for trials lasting at least one month (SMD = −0.28, [−0.44, −0.13]) [104] Significant difference in depression score (SMD = −0.47 [−0.67, −0.27]) [105] Other meta-analyses reveal no significant difference, though very close to statistical threshold of $p$ = 0.05 (SMD = −0.128, [−0.261, 0.005]) [106] | Biomarkers are multifactorial [107] | 10 billion CFU [108] | Yogurts, kefir [109], kombucha [110], fermented meat and fish products, sauerkraut, kimchi, natto, miso, sourdough bread [111,112] |
| Acetyl-L-Carnitine | Significant reduction in depressive symptoms (SMD = −1.10, [−1.65, −0.56]) [113] | 10–15 µmol/L [114] | 2 g [115] | Meat [116] |
| Creatine | Reduction in depression associated with level of dietary creatine consumption AOR = 0.68 [0.52, 0.88] [117] | N/A | 2–10 g [118] | Meat [119] |

**Table 1.** *Cont.*

| Factor | Impact | Optimal Serum Levels | Daily Intake in Depression Treating Context | Sources |
|---|---|---|---|---|
| Amino acids (a.a.) | Reduction in depressive symptoms greater than placebo SMD = −1.21 [0.57, 1.95] [120] | Varies by specific a.a. For tryptophan: 40–120 µmol/L [121] | Recommended Daily Allowance (RDA) doses of 8 essential and 2 semi-essential amino acids (arginine and histidine) [122] | Protein rich foods, supplements |
| Methylfolate | Improvement in depression profile SMD = −0.38 [−0.59, −0.17] [123] SMD = −0.61 [−0.97, −0.24] [124] | Serum 5-methyltetrahydrofolate 24–51nmol/L [125] | 15 mg [124] | Leafy greens, legumes, fortified cereals, liver |
| 5-HTP | Significant improvements in depression symptoms (g = 1.11 [0.53, 1.69] [126] | N/A | 150–300 mg [126] | Turkey, chicken, fish, dairy products, supplements |
| St. John's Wort | Similar response to SSRI treatment. RR = 0.96 [0.83, 1.10] relative to second generation antidepressants [127] | N/A | 500 mg [128] | *Hypericum perforatum* |
| Saffron | Significantly better than placebo improvement in depressive symptoms g = 0.891 [0.369, −1.412] [129] | N/A | 100 mg [129] | Saffron spice derived from the *Crocus sativus* flower |
| Curcumin | Significant clinical efficacy in depression (HAM-D SMD = −0.34 [−0.56 to −0.13]) [130] Effective as adjunctive therapy [131] | N/A | 1 g [130] | Turmeric spice, commonly used in curry dishes and various recipes |
| Methylene Blue | Reduction in manic depressive attacks [132] Marked improvement in depressive symptoms SMD = −0.99 [−1.82, −0.16] [133] | N/A | 15 mg/day | Supplements |
| Chinese Herbal Medicine | Positive effect [134,135] CHM better than placebo (HAMD-17, MD = −4.53, [−5.69, −3.37]) [134] | N/A | N/A | Depends on formulation |
| Nigella Sativa | Decreased depression score SMD = −1.4 [−1.94, −0.86] [136] | N/A | 1000 mg oil extract | Black cumin seed |
| S-adenosyl methionine | Low-quality evidence for efficacy [137] | N/A | 1600 mg orally [138] | Supplements |
| Bacopa Monnieri | Nonsignificant improvement SMD = −0.32 [−0.86, 0.22] [139] | N/A | 300 mg extract [139] | Bacopa Monnieri |

**Table 1.** *Cont.*

| Factor | Impact | Optimal Serum Levels | Daily Intake in Depression Treating Context | Sources |
|---|---|---|---|---|
| SHR-5 (Rhodiola metabolite) | Improves depressive symptoms SMD = −1.66 [−2.17, −1.16] [140] | N/A | 340−680 mg Rhodiola [140] | *Rhodiola rosea* L. |
| Kava kava | Improvement in symptoms in human subjects SMD = 2.24 ($p < 0.0001$) [141] | N/A | 3.2 g [141] | Piper methysticum |
| Inositol | Equivocal evidence [68] | 7 µg/mL [142] | 12 g [143] | Fruits, beans, grains, and nuts [144] |
| Chromium | RCT shows effectiveness compared to placebo nonsignificant SMD = −0.538 [−1.72, 0.65] [145] | <0.60 µg/L [146] | 600 µg chromium picolinate [145] | Meats, grain products, fruits, vegetables, nuts, spices, brewer's yeast, beer, and wine [147] |
| Co-enzyme Q10 | SMD = 0.97 [0.01, 1.93] $p < 0.00001$ [69] | Males: 0.9 µmol/L Females: 0.8 µmol/L [148] | 300 mg [69] | Meat, fish, nuts, and some oils [149] |
| Crocin | SMD = 6.04 [3.43, 8.65] $p = 0.01$ [69] | N/A | 30 mg [69] | Saffron |
| Antioxidant supplements | Significant improvement (SMD = 0.40, 95% CI = 0.28–0.51, $p < 0.00001$) [69] | N/A | N/A | Supplements |
| Extra Virgin Olive Oil | Antidepressant activity in severely depressed patients SMD = −0.75 [−1.23, −0.27] [150] | N/A | 25 mL extra virgin olive oil [150] | Extra virgin olive oil |
| Lavender | Positive impact of lavender with imipramine (antidepressant) compared to imipramine monotherapy SMD = 2.45 [1.67, 3.23] [78] | N/A | 60 drops lavandula tincture [151] | *Lavandula angustifolia* |
| Dan zhi xiao yao | Decrease in Self-Rating Depression Scale scores [WMD = 0.89, 95% CI (−6.33, 8.11); $p = 0.81$] [152] | N/A | 24 g [152] | Mixture of *Bupleurum chinense, Scutellaria baicalensis, Paeonia lactiflora, Glycyrrhiza uralensis, Mentha haplocalyx, Zingiber officinale,* and *Ziziphus jujuba* |
| Alpha Lipoic Acid | Equivocal evidence [68] | N/A | 600–1800 mg [153] | Muscle meats, heart, kidney, and liver [154] |
| N-acetyl Cysteine (NAC) | Positive evidence from trials [155,156] | N/A | 1 g [156] | Supplements |
| Ginseng | Improvements in QOL in patients complaining of stressor fatigue [157] | N/A | 17.4 mg Panax Ginseng extract with a blend of multivitamins [157] | Ginseng |

Other agents with preclinical data are shown in Supplementary Table S2 [141,158–218]. Preclinical studies typically focus on several behavioral tests in mice.

*4.1. The Gut Microbiome and Depression: The Importance of the Gut–Brain Axis*

Depression is a complex mental health disorder that affects millions worldwide. While the exact causes of depression are not fully understood, emerging research suggests that the gut microbiome may play a significant role in its development and progression. The gut–brain axis, a bidirectional communication system between the enteric microbiota and the central nervous system, is thought to be a key mediator in this relationship and seems to have significant implications for depression.

### 4.1.1. The Gut–Brain Axis

The gut–brain axis refers to the intricate interactions between the enteric microbiota, the central nervous system (CNS), and the enteric nervous system (ENS) [219], creating a paradigm change in neuroscience [220]. The enteric microbiota, consisting of trillions of microorganisms residing in the gastrointestinal tract, influences various physiological processes, including the immune function, metabolism, and neurotransmitter production [221]. These microorganisms produce neurotransmitters, such as serotonin and gamma-aminobutyric acid (GABA), which are known to regulate mood and emotions [222] and even modify epigenetic processes of the gut–brain axis [223].

### 4.1.2. The Role of the Gut Microbiome in Depression

Studies have found alterations in the composition and diversity of the gut microbiome in individuals with depression [224]. Researchers [225] highlighted the bidirectional communication between the gut microbiota and the CNS, emphasizing the gut microbiome's influence on neurological and psychiatric disorders [226]. Additionally, others [227] discussed the impact of the gut–brain axis on mental health, emphasizing the potential therapeutic benefits of modulating the gut microbiota.

### 4.1.3. Mechanisms

Several mechanisms have been proposed to explain how the gut microbiome may contribute to depression [222]. Short-chain fatty acids (SCFAs) [228], produced by the gut microbiota through the fermentation of dietary fibers, have been shown to modulate brain function and behavior [229]. Scientists [230] discussed the role of SCFAs in microbiota–gut–brain communication, highlighting their potential as therapeutic targets. Moreover, a dysregulated microbiota–gut–brain axis has been observed in patients with bipolar depression [231,232] and associated with depressive-like behaviors in animal models [224,226]. Emerging evidence suggests that alterations in gut permeability [233] and the subsequent inflammatory response may play a crucial role in the relationship between the gut microbiome and depression [234]. Research demonstrated [221] how the gut microbiome influences the production and metabolism of neurotransmitters such as serotonin [235], dopamine [236], and gamma-aminobutyric acid (GABA) [237], and how alterations in these neurotransmitter systems may contribute to depressive symptoms.

### 4.1.4. Clinical Implications and Treatment Approaches

Understanding the gut–brain axis and its association with depression opens up possibilities for novel therapeutic interventions. In studies, a predominance of some potentially harmful bacterial groups or a reduction in beneficial bacteria [232] has been found in depressive patients. Dietary interventions have been the subject of research and studies examining their potential impact on symptoms of depression. There is emerging evidence that suggests a link between diet and mental well-being, indicating that dietary improvements may positively affect symptoms of depression [71,238]. Probiotics, which are live microorganisms that confer health benefits when consumed, have shown promise in modulating the gut microbiota and improving depressive symptoms. Research [239] reviewed the mechanisms of action of probiotics as potential therapeutic targets for depression and anxiety disorders. For example, Lactobacillus rhamnosus directly regulates the GABAergic system in a vagus nerve-dependent way and mitigates depression- and anxiety-like behaviors in mice [240].

Bifidobacterium breve, proven to have an antidepressant-like effect, could stimulate the production of intestinal 5-hydroxytryptophan in mice and then regulate the host's serotonin metabolism [241,242]. Pediococcus acidilactici could mitigate anxiety symptoms in mice by producing lactic acid and inhibiting the over-proliferated gut pathogenic bacteria under stress [243]. Fecal Microbiota Transplantation (FMT) is a procedure in which a healthy donor's fecal matter is transplanted into a recipient's gastrointestinal tract to restore a healthy balance of gut bacteria. There is growing interest in the potential therapeutic effects of FMT on various conditions, including depression [244,245].

Overall, the gut microbiome and the gut–brain axis are emerging areas of research in the field of depression. The bidirectional communication between the gut microbiota and the CNS highlights the potential for microbiome-based interventions in treating depression. While promising, more research is required to elucidate the underlying mechanisms and develop targeted therapies to modulate the gut–brain axis to alleviate depressive symptoms effectively.

### 4.2. The Link between Depression and Inflammation

Depression, a prevalent mental health disorder, has long been associated with alterations in the immune system and chronic inflammation. The findings highlight potential therapeutic targets and the importance of a holistic approach to managing depression. The etiology of depression remains multifactorial and complex; emerging evidence suggests a strong connection between depression and inflammation [246]. Inflammation, traditionally associated with the immune response to infection or injury, has been implicated in the pathophysiology of various psychiatric disorders. Numerous studies have demonstrated elevated levels of pro-inflammatory cytokines [247], such as interleukin-6 (IL-6) [248,249] and tumor necrosis factor-alpha (TNF-$\alpha$) [250], in individuals with depression. Conversely, chronic inflammation, often triggered by external factors such as stress, trauma, or medical conditions, has been shown to contribute to developing or exacerbating depressive symptoms. The dysregulation of the immune system, particularly the imbalance in pro-inflammatory and anti-inflammatory cytokines, plays a crucial role in altering neurotransmitter metabolism, neuroplasticity, and the neuroendocrine function, ultimately affecting mood regulation [251].

The bidirectional relationship between depression and inflammation suggests a complex interplay between the immune and central nervous systems. Inflammation-induced activation of the kynurenine pathway [252], dysregulation of the hypothalamic–pituitary–adrenal (HPA) axis [253], and disruption of the blood–brain barrier [254] are among the proposed mechanisms linking inflammation to depressive symptoms. Moreover, chronic inflammation may impair the efficacy of conventional antidepressant treatments, emphasizing the need for personalized approaches that target both the neurochemical imbalances and the underlying inflammatory processes. It is worth noting that studies indicate that EMF exposure can increase the secretion of pro-inflammatory cytokines [255], including IL-6, TNF-alpha, and IL-1. The increasing intensity of EMF via mobile phones, Wi-Fi, etc., should initiate more research into the potential association between depression and EMF devices. This pro-inflammatory effect has been shown to be inhibited by curcumin [256].

Curcumin, a compound found in turmeric, has been the subject of scientific research exploring its potential use in depression [130,257–259]. Curcumin has been found to possess anti-inflammatory properties, which, as we can see, may be relevant to depression. Inflammation has been implicated in the development and progression of depression, and curcumin's anti-inflammatory effects may help alleviate depressive symptoms [260]. Curcumin has also shown neuroprotective properties in preclinical studies, including antioxidant and anti-apoptotic effects. These effects may help protect against neuronal damage and promote neuroplasticity, essential factors in depression [131]. It has been found to modulate various neurotransmitters, including serotonin, dopamine, and glutamate, which are involved in mood regulation. By influencing these neurotransmitter systems, curcumin may impact depressive symptoms [260]. BDNF is a protein that plays a crucial

role in the growth and maintenance of neurons. Reduced levels of BDNF have been associated with depression. Curcumin has been shown to increase BDNF levels, possibly contributing to its potential antidepressant effects [261]. Some studies have explored the combination of curcumin with other antidepressant medications, suggesting possible synergistic effects. Combining curcumin with standard antidepressant treatment may enhance the therapeutic response [258,259].

*4.3. The Complex Relationship between Thyroid Dysfunction and Depression*

Thyroid dysfunction refers to the abnormal functioning of the thyroid gland, which can result in either hyperthyroidism (overactive thyroid) or hypothyroidism (underactive thyroid). Depression, on the other hand, is a mood disorder characterized by persistent feelings of sadness, loss of interest, and a lack of motivation. While the connection between thyroid dysfunction and depression has been the subject of scientific inquiry [262–264], the relationship between these two conditions remains complex and multifaceted [265]. Research has shown a bidirectional relationship between thyroid dysfunction and depression, with each condition potentially influencing the other [266]. Several studies have found that individuals with thyroid dysfunction are at a higher risk of developing depression. For instance, a meta-analysis found a significant association between hypothyroidism and depression [267], suggesting that individuals with an underactive thyroid may be more prone to depressive symptoms. Moreover, thyroid hormones play a crucial role in regulating neurotransmitters [268] such as serotonin [235], dopamine [269], and norepinephrine [270], which are involved in mood regulation. Imbalances in these neurotransmitters have been linked to the development of depression. Therefore, disruptions in thyroid hormone levels can impact the functioning of these neurotransmitters, potentially contributing to the development of depressive symptoms [271]. Conversely, depression may also affect thyroid function. Chronic stress, a common contributor to depression, can lead to dysregulation of the hypothalamic–pituitary–thyroid (HPT) axis [272] which controls thyroid hormone production. This dysregulation can result in alterations in thyroid hormone levels [273,274], potentially leading to thyroid dysfunction. Autoimmune thyroiditis is also associated with an increased risk of depression [275]. Elevated thyroid-stimulating hormone (TSH), antithyroglobulin (TgAb), and thyroid peroxidase antibodies (TPOAb) levels have all been linked to depression and an increased risk of suicide [266]. Moreover, hypothyroidism is known to be one of the leading causes of treatment-resistant depression. Furthermore, chronic inflammation, often observed in individuals with depression, can also impact thyroid function. The complex relationship between thyroid dysfunction and depression necessitates comprehensive treatment approaches that address both conditions. For individuals with thyroid dysfunction, appropriate thyroid hormone replacement therapy can help restore hormonal balance and alleviate depressive symptoms. It is crucial to closely monitor thyroid hormone levels and micronutrients, such as iodine, zinc [276], iron [277], and selenium [278], and adjust medication dosages as necessary. If an individual with depression also exhibits symptoms of thyroid dysfunction, it is important to assess thyroid function and consider appropriate interventions to optimize treatment outcomes [279]. Thyroid dysfunction and depression share a complex and bidirectional relationship. While individuals with thyroid dysfunction may be at a higher risk of developing depression [280], depression can also impact thyroid function [273]. Addressing both conditions simultaneously is crucial for effective treatment outcomes. Further research is needed to unravel the precise mechanisms underlying this relationship and develop targeted interventions that can improve the lives of individuals affected by both thyroid dysfunction and depression.

## 5. Lifestyle Changes for Treatment of Depression

There are several changes that one can make in one's life to recover from depression. These useful strategies have an evidence base documenting their efficacy. We performed a literature search on lifestyle treatment for depression and found several reviews (Supplementary Table S3) [281–289]. These findings are summarized below in Table 2.

**Table 2.** A summary of lifestyle interventions and their impacts on depression.

| Intervention | Effect |
|---|---|
| Dance | Antidepressant impact (SMD = 0.50, *p* = 0.01) [290] |
| Mindfulness | Decreases in depressive symptoms (SMD = 0.31–0.56) [291,292] |
| Sleep | Improved sleep quality decreases depressive symptoms (SMD = −0.45 [−0.55, −0.36]) [293] |
| Natural environments | Increases positive mood and lowers feelings of depression SMD = −0.67 [−0.99, −0.35] [294] |
| Time with animals | Reduction in depressive symptoms (SMD = 0.61 [0.03, 1.19]) [295]. |
| Socialization | Significant improvement in depressive scores SMD = 0.18 [−0.00 to 0.36] [296] |
| Journaling | Positive impact (SMD = 0.61 [0.19, 1.02]) [297] |
| Gratitude | Associated with positive mental health, including alleviating depression (SMD = Reduction in depressive symptoms 0.29 [−0.37, −0.23]) [298] |
| Deep brain stimulation | Reduction in mean depression score SMD = –0.42 [–0.72, −0.12] [299] |
| Sauna/whole body hyperthermia | Reduced odds of depressive symptoms for people using sauna OR = 0.60 [0.39, 0.90] [300] |

*5.1. Exercise*

There is growing recognition that lifestyle behaviors, such as physical activity and exercise, can be useful strategies for treating depression, reducing depressive symptoms, improving quality of life, and improving physical health outcomes. Cross-sectional studies have shown that people with higher levels of physical activity present decreased depressive symptoms, and these results are consistent across different countries and cultures. For example, recent evidence using data from the Brazilian National Health Survey, accounting for 59,399 individuals, demonstrated that a lack of physical activity for leisure was associated with depression in young males, and middle-aged and older adults [301]. A study across 36 countries demonstrated that lower levels of physical activity (defined as less than 150 min of moderate–vigorous physical activity per week) were consistently associated with elevated depression (OR, 1.42; 95%CI, 1.24–1.63) [302]. However, mental health benefits have been noted from being physically active, even at levels below the public health recommendations [303]. In The Irish Longitudinal Study on Ageing, participants performing 400 to less than 600 MET-min/wk had a 16% lower rate of depressive symptoms (adjusted incidence rate ratio [AIRR], 0.84; 95% CI, 0.81–0.86) and 43% lower odds of depression compared with 0 MET-min/wk [304]. These findings are consistent with recent meta-analytic data suggesting that salutary mental health benefits among adults can be achieved with physical activity below public health recommendations; specifically, an activity volume equivalent to 2.5 h per week of brisk walking was associated with a 25% lower risk of depression, and half that activity volume was associated with an 18% lower risk compared with no activity [303]. The findings of The Irish Longitudinal Study on Ageing suggest that accumulating as little as 100 min per week or 20 min per day for 5 days per week of moderate-intensity activity (e.g., brisk walking; 4 METs) may be sufficient to significantly lower the risk of depressive symptoms and odds of major depression over time among older adults.

A large body of trials has been performed over the last 40 years evaluating the role of exercise as a therapy for depression. These results have been summarized in several meta-analyses. In a Cochrane analysis of 35 trials (1356 participants) comparing exercise

with no treatment or a control intervention, the pooled outcome for the primary outcome of depression at the end of treatment was a standardized mean difference (SMD) of −0.62; 95% CI −0.81 to −0.42, indicating a moderate clinical effect. Schuch et al. performed a meta-analysis which included 25 RCTs comparing exercise versus control comparison groups. [305] Overall, exercise had a large and significant effect on depression. Similarly, Krogh et al. performed a meta-analysis which included 35 trials enrolling 2498 participants [306]. The effect of exercise versus control on depression severity was −0.66 SMD [95% CI −0.86 to −0.46; $p < 0.001$].

Exercise can improve depressive symptoms in people with depression. However, like other treatments, exercise is not a panacea and may not work equally for all. A seminal study by Dunn et al. named "*The Depression Outcomes Study of Exercise*" found a response rate of about 40% in depressed people free from other treatments [307]. However, it is likely that when combined with other interventions (i.e., vitamin D, L-methyl-folate, etc.) the response rate and degree of response will be much greater. In essence, exercise has multiple benefits to several domains of physical and mental health and should be promoted to everyone. To ensure compliance, adapting exercise prescriptions for people with depression should account for personal preferences and previous experiences in terms of making it the most enjoyable experience possible. Acute exercise should be used as a symptom management tool to improve mood in depression, with even light exercise an effective recommendation [308]. These data suggest that physical activity is beneficial for the depressed patient regardless of the intensity of the exercise.

The neurobiological mechanisms underpinning the antidepressant effects of exercise are largely unclear. However, some hypotheses involving inflammation, oxidative stress, and neuronal regeneration are speculated. Exercise training can promote increases in anti-inflammatory and antioxidant enzymes, referred to as a hormesis response, and subsequently decrease IL-6 levels. This effect was demonstrated in the REGASSA trial, where decreases in IL-6 serum levels were associated with reductions in depressive symptoms [309].

### *5.2. Time in Nature*

Time in nature is associated with increases in positive mood and lowered feelings of depression [310,311].

### Animal-Assisted Therapy

Time spent with animals can be an effective way of reducing depression [295].

### *5.3. Mindfulness*

Several mindfulness-based therapies can potently treat depression. The most studied treatments are cognitive behavioral therapy (CBT), mindfulness-based stress reduction (MBSR), and mindfulness-based cognitive therapy (MBCT), which have important distinctions. Mindfulness-based therapies demonstrate significant reductions in depression [292].

### *5.4. Connection with Others*

In the middle of depression, some of the things that fall by the wayside are plans and social interactions. Existing in large cities, one lives a largely anonymized existence, where one does not experience connection with others, including seeing others and being seen by others.

### Purpose and Goals

Positive, goal-directed activity is associated with a decrease in depressive symptoms and has the added benefit of providing structure and a reason to positively interact and create with others. Progress in any aspect increases positive self-regard, confidence, and a sense of self-efficacy, as well as one's social status. These factors are associated with a decrease in depressive symptoms [312,313].

Another benefit is that learning positively uses neural pathways and grows new neurons and is also associated with a sense of optimism. Furthermore, a challenging task necessarily takes much of the mental bandwidth, leaving less space for ruminations characteristic of depression. During periods of intense stress including the London Blitz during World War II, there was a paradoxical decrease in psychiatric presentation to hospitals, owing to the dire need of hospital beds [314]. The efforts of every man and woman were needed, and this sense of purpose is protective against depression.

Depression is often a reason for introspection into the aspects of one's life that are not working. Often a major life area, such as one's career or one's close relationships (or lack thereof), is brought into focus. In these cases where dissatisfaction with one's current life is the proximal cause of one's depression, working with a life coach or otherwise reflecting on one's ideal life (and how to achieve it) is a powerful practice for inspiring hope and action which follows that. A significant proportion of depression is a lack of meaning and purpose.

Interestingly, regular Argentinian tango was comparable to mindfulness meditation in terms of the impact on depression [290], suggesting a value in novel pursuits and hobbies.

Indigenous communities living traditional existences do not suffer from psychiatric issues. The differences in the sense of purpose between modern and tribal cultures can be attributed to the tight-knit tribal communities where everybody feels a sense of importance in the eyes of the community. Furthermore, practices such as initiation into manhood and womanhood, most notably the vision quest, provide the individual with a clear role to play in the community.

Over millennia, these practices have corrected wayward youth and integrated at-risk youth into constructive roles within the community. While this review mostly focuses on the individual treatment of depression, it should be noted that initiatives like Upward Bound, which provide a similar experience for youth on the cusp of adulthood, increase the likelihood of post-secondary education [315].

These programs involve people getting out in nature, which in itself has positive benefits for mood disorders [316], and additionally provides the benefits of physical activity [317]. The program positively impacts self-concept [318].

### 5.5. Gratitude

An outlook of gratitude has been valued by all the major monotheistic religions [319]. Furthermore, in modernity, when gratitude is operationalized as an explicit practice, it is associated with positive mental health, including alleviating depression [298,320–323]. Gratitude journaling, which simply involves recording 3–5 things that one is grateful for daily, is one of the most accessible ways that one can practice journaling [324,325].

### 5.6. Deep Brain Stimulation

Noninvasive brain stimulation methods have been studied for their favorable modulation of a wide variety of neural states. For the treatment of depression, some promising data exist for these therapies, showing a small but significant effect [298,321].

Non-invasive brain stimulation (NIBS) using transcranial direct current stimulation or transcranial magnetic stimulation has been demonstrated to be highly effective in the treatment of depression [326–330].

### 5.7. Whole-Body Hyperthermia

Historically, hyperthermia interventions have been utilized to address depressive symptoms, with evidence dating back to ancient times, such as the practices of Galen of Pergamon (129–198 C.E.), who treated melancholia by immersing patients in hot tubs and providing skin massages [331]. Contemporary research has highlighted the positive effects of regular sauna bathing, including reductions in all-cause and cardiovascular mortality, increased lifespan, improved exercise performance, and the activation of autophagy through the expression of heat-shock proteins [332–335]. Heat therapy also enhances cell

stress pathways, possesses antioxidant and anti-inflammatory properties, and enhances mitochondrial function. Sauna bathing exhibits physiological similarities to aerobic exercise, increasing heart rate, stroke volume, and cardiac output [336,337]. Furthermore, whole-body hyperthermia (WBH) selectively raises IL-6 levels [338] and shows promise in conditions like chronic fatigue syndrome [339,340].

Animal studies have indicated that WBH activates portions of the dorsal raphe nucleus associated with mood regulation and produces antidepressant-like responses [341]. Clinical studies have shown that a single session of WBH can significantly reduce depressive symptoms when assessed five days post-treatment [342]. Additionally, a randomized, double-blind study comparing WBH with a sham condition in depressed patients revealed significant reductions in Depression Rating Scale scores over a six-week post-intervention period in the active WBH group [343]. Hanusch et al. conducted a meta-analysis on the effect of WBH on depression indices, encompassing seven studies with a total of 148 subjects. Six out of seven studies reported statistically significant reductions in depressive symptoms between one and six weeks post-intervention. The treatment effect appeared to be independent of the total number of WBH sessions, with target temperatures between 38 °C and 39 °C and a slower increase in core body temperature during the intervention resulting in larger treatment effects. This suggests potential benefits of a near-infrared (NIR) sauna over a regular sauna, as NIR sauna sessions can be more controlled, shorter in duration (5–10 min initially, increasing to 20 min), and performed two to three times a week for maximal cardiovascular benefit [331].

### 5.8. Photobiomodulation

Photobiomodulation (PBM) is referred to in the literature as low-level light therapy, red-light therapy, and near-infrared (NIR)-light therapy. Depression is associated with brain hypometabolism and cerebral as well as systemic mitochondrial dysfunction [344–356]. In a rat model of depression, vital steps in the production of adenosine triphosphate (ATP) were inhibited in the cerebral cortex and cerebellum [347].

Peripheral blood mononuclear cells of depressed patients were shown to have significantly impaired mitochondrial function [348,349], and greater mitochondrial dysfunction correlated with the severity of neuro-vegetative symptoms, including fatigue and poor concentration [348]. Muscle biopsy samples from depressed patients with physical symptoms had a decreased rate of ATP production and more frequent mitochondrial DNA deletions than controls [346].

The most well-studied mechanism of action of PBM centers around enhancing the activity of cytochrome C oxidase (CCO), which is unit four of the mitochondrial respiratory chain, responsible for the final reduction of oxygen to water [350]. The theory is that CCO enzyme activity may be inhibited by nitric oxide (NO). This inhibitory NO can be dissociated by photons of light that are absorbed by CCO. These absorption peaks are mainly in the red (600–700 nm) and near-infrared (760–940 nm) spectral regions. When NO is dissociated, the mitochondrial membrane potential is increased, more oxygen is consumed, more glucose is metabolized and more ATP is produced by the mitochondria [351].

PBM has been found to specifically increase CCO activity and expression [350,352,353]. Studies have also shown increases in complex II, III, and IV activity, as well as upregulation of gene coding for subunits of complex I, complex IV, and ATP synthase. [350] Low-level laser therapy has been shown to increase levels of ATP, the rate of oxygen consumption, and cerebral oxygenation [350]. Though t-PBM with red and NIR light can include wavelengths from 600 to 1070 nm, specific wavelengths have been directly linked to mitochondrial activity. Near Infrared activates CCO, increases mitochondrial oxygen consumption, and leads to higher levels of ATP [354–356].

There is some evidence that PBM applied peripherally, not just transcranially, may have an effect in attenuating depressive symptoms [350]. There is no clear mechanism proposed explaining this effect. In a recent study, five outpatients with lower back pain and concurrent self-reported depression were treated over five weeks with physical therapy

(PT) (5 sessions) and concurrent PBM (3 sessions) and matched to five control patients treated with PT alone (5 sessions) [357]. Participants receiving s-PBM reported a larger decrease in their depression scores. Oron and co-workers have shown that delivering NIR light to mouse tibia resulted in improvement in a transgenic mouse model of Alzheimer's disease [358].

## 6. Conclusions

More integrative approaches, including diet and lifestyle, may improve the quality of life of people with depression and enable them to live a fulfilling life. Currently, practitioner understanding is a barrier, as well as the limited time that primary care physicians spend with patients. Additionally, other systemic issues remain regarding the cost of therapy, while simultaneously therapists themselves are overburdened and under-compensated. Expectations of one's self can be a barrier to those experiencing depression even in acknowledging it, let alone seeking help [359]. Much work remains to be carried out on the public health understanding of depression, and an integrative approach, combining dietary and lifestyle change with other therapies and embracing a trauma-aware perspective, can help greatly. Additionally, the resiliency of the person experiencing depression must be acknowledged, and he or she must be captain of the process. Practitioners can help by educating and coaching the person recovering from depression, and by pointing them to resources and therapeutics for their specific case.

**Supplementary Materials:** The following supporting information can be downloaded at: https://www.mdpi.com/article/10.3390/psycholint6020033/s1, Table S1: Dietary, nutraceutical and herbal interventions for treating depression in human subjects found in reviews of dietary interventions for depression. Table S2: Dietary, nutraceutical and herbal interventions for treating depression in preclinical models found in reviews of dietary interventions for depression. Table S3: Lifestyle interventions for treating depression in humans found in reviews of lifestyle interventions for depression.

**Author Contributions:** Conceptualization, P.E.M.; methodology, M.H. and P.E.M.; investigation, M.H., C.P. and P.E.M.; data curation, M.H.; writing—original draft preparation, M.H., C.P. and P.E.M.; writing—review and editing, M.H., C.P. and P.E.M.; supervision, P.E.M.; funding acquisition, P.E.M. All authors have read and agreed to the published version of the manuscript.

**Funding:** This research was funded by the Frontline COVID-19 Critical Care Alliance and The APC was funded by the Frontline COVID-19 Critical Care Alliance.

**Acknowledgments:** We thank Mazhar Hussain for assisting with reference formatting.

**Conflicts of Interest:** The authors declare no conflicts of interest.

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
