# Peer review of "Integrative Interventions for Improving Outcomes in Depression: A Narrative Review"

_2813-9844, doi:10.3390/psycholint6020033_

Round 1

Reviewer 1 Report

The paper describes many different factors possibly influencing depression i.e. diet, lifestyle and psychedelic substances. Much of the information provided needs to be clarified

The work is chaotic, I would suggest to clearly separate the chapters that describe risk factors/protecting factors of depression from those affecting the course of disease/treatment.

 It suggest to provide more details about the discussed substances/effects e.g. doses tested, time of administration, groups tested. Does it make sense to take zinc/magnesium/calcium/vitamins etc. if the levels of these substances in the body are normal?

In the case of most substances analyzed, usually one citation supporting the assumed thesis is provided, this is not enough and indicates that the analysis may be biased.

It is not clear why the discussion about psychedelics was included? It could be concluded that taking psychedelic substances or vitamins/minerals is essentially the same thing. I suggest to remove this part from the MS.

no detail comments

Author Response

Reviewer 1: The paper describes many different factors possibly influencing depression i.e. diet, lifestyle and psychedelic substances. Much of the information provided needs to be clarified

We thank the reviewer for his or her comments and we have removed some of the extraneous information (psychedelic treatment of depression) for greater consistency in information.

The work is chaotic, I would suggest to clearly separate the chapters that describe risk facstors/protecting factors of depression from those affecting the course of disease/treatment.

This current work now focuses on interventional trials (i.e. for treatment, not prevention of depression). We do discuss factors associated with depression, in the section on epidemiology, which relates to prevention.

It suggest to provide more details about the discussed substances/effects e.g. doses tested, time of administration, groups tested.

For the nutritive interventions, we have added the daily dose in the intervention group. 

Does it make sense to take zinc/magnesium/calcium/vitamins etc. if the levels of these substances in the body are normal?

We have added reference ranges for serum levels of different vitamins and minerals where applicable. We have also added the interventional dose of the substance in question.

In the case of most substances analyzed, usually one citation supporting the assumed thesis is provided, this is not enough and indicates that the analysis may be biased.

We have striven to find the most recent and comprehensive meta analysis on the intervention used in interventional trials. We have reported the Standardized mean difference with confidence intervals for each value where applicable.

It is not clear why the discussion about psychedelics was included? It could be concluded that taking psychedelic substances or vitamins/minerals is essentially the same thing. I suggest to remove this part from the MS.

We have removed the part on psychedelics from the manuscript. 

We thank reviewer 1 for his or her comments.

Author Response

We thank the reviewer for his or her input.

The table and section callouts in the aims and methods have been corrected. 

Reviewer 3 Report

This is a very interesting and comprehensive paper. Use more than 300 literatures’ review to display more than 100 food supplements and  lifestyles that affected depression,  due to the content is too broad and difficult to focus, the reviewer suggests the authors may reorganize the paper, or divide the current paper into two articles in addition to apply  evidence-based or systemic literature review way of writing, not only to list all the papers outcome, it needs a better scientific way to present those papers before to the readers 

Table 1-3 all need to be revised, and the content also will suggest to use the systemic literature review to write up. 

Author Response

This is a very interesting and comprehensive paper. Use more than 300 literatures’ review to display more than 100 food supplements and lifestyles that affected depression, due to the content is too broad and difficult to focus, the reviewer suggests the authors may reorganize the paper, or divide the current paper into two articles in addition to apply evidence-based or systemic literature review way of writing, not only to list all the papers outcome, it needs a better scientific way to present those papers before to the readers.

Regarding the division of the article,  we have decided to remove the section on the psychedelic treatment of depression, as this is the least relevant section to the manuscript. 

Regarding presenting conclusions in a more scientific manner, we have reported the standardized mean difference for the trials, with confidence ranges, for comparison of different interventions. We have also included the therapeutic dose for each nutritive agent to allow the reader to grasp the dosages.

 Table 1-3 all need to be revised, and the content also will suggest to use the systemic literature review to write up.

We have removed the table on the psychedelic treatment of depression. For the remaining tables, we report interventional studies in a more consistent format where dosages and effect sizes, as standardized mean differences with confidence intervals. 

We thank the reviewer for his or her comments and we hope that we have adequately addressed his or her concerns. 

Round 2

Reviewer 2 Report

The modification makes it easier to understand. 

The title is appropriate for your article. The results are clearly presented.

Reviewer 3 Report

Depression has had caused the 21st-century great impact on humans, it is very important to reveal how many possible ways to cope with it and whether they work. This is a paper to summarize the literature on non-invasive dietary and lifestyle approaches for treating depression. The authors have searched for reviews on nutritional supplements in depression and found some with an evidentiary overview of different nutraceutical and nutritional supplementary protocols for depression. In addition, a great amount of work has been done to draw an overall picture of current depression integrative treatments. 

However, a minor suggestion for the authors in the future is that the content can use a systemic reviewing and ranking to provide an evidence-based perspective view and give the reader scientific confidence for further practice. Especially on Table 1, and Table 2. To suggest the authors add a co-review system to up-grade the reliability.

A minor suggestion for the authors in the future is that the content can use a systemic reviewing and ranking to provide an evidence-based perspective view and give the reader scientific confidence for further practice. Especially on Table 1, and Table 2. To suggest the authors add a co-review system to up-grade the reliability.